# Shock Index for Early Detection of Low Plasma Fibrinogen in Trauma: A Prospective Observational Cohort Pilot Study

**DOI:** 10.3390/jcm12041707

**Published:** 2023-02-20

**Authors:** Josef Škola, Marcela Bílská, Michala Horáková, Václav Tégl, Jan Beneš, Roman Škulec, Vladimír Černý

**Affiliations:** 1Department of Anaesthesiology, Perioperative Medicine and Intensive Care, J. E. Purkinje University, Masaryk Hospital in Usti nad Labem, Socialni Pece 3316/12A, 401 13 Usti nad Labem, Czech Republic; 2Faculty of Medicine in Hradec Kralove, Charles University, Šimkova 870, 500 03 Hradec Kralove, Czech Republic; 3Department of Anaesthesiology, Resuscitation and Intensive Care, Faculty of Medicine in Pilsen, Charles University, Alej Svobody 80, 304 60 Plzen-Lochotin, Czech Republic; 4Department of Anaesthesiology, Resuscitation and Intensive Care, University Hospital in Pilsen, Alej Svobody 80, 304 60 Plzen-Lochotin, Czech Republic

**Keywords:** shock index, trauma, coagulopathy, hypofibrinogenemia, fibrinogen concentrate

## Abstract

Shock index (a ratio between heart rate and systolic blood pressure) predicts transfusion requirements and the need for haemostatic resuscitation in severe trauma patients. In the present study, we aimed to determine whether prehospital and on-admission shock index values can be used to predict low plasma fibrinogen in trauma patients. Between January 2016 and February 2017, trauma patients admitted from the helicopter emergency medical service into two large trauma centres in the Czech Republic were prospectively assessed for demographic, laboratory and trauma-associated variables and shock index at scene, during transport and at admission to the emergency department. Hypofibrinogenemia defined as fibrinogen plasma level of 1.5 g·L^−l^ was deemed as a cut-off for further analysis. Three hundred and twenty-two patients were screened for eligibility. Of these, 264 (83%) were included for further analysis. The hypofibrinogenemia was predicted by the worst prehospital shock index with the area under the receiver operating characteristics curve (AUROC) of 0.79 (95% CI 0.64–0.91) and by the admission shock index with AUROC of 0.79 (95% CI 0.66–0.91). For predicting hypofibrinogenemia, the prehospital shock index ≥ 1 has 0.5 sensitivity (95% CI 0.19–0.81), 0.88 specificity (95% CI 0.83–0.92) and a negative predictive value of 0.98 (0.96–0.99). The shock index may help to identify trauma patients at risk of hypofibrinogenemia early in the prehospital course.

## 1. Introduction

Trauma is a significant global public health issue causing 10% of all deaths worldwide [1,2,3]. Uncontrolled haemorrhage and severe traumatic brain injury are the most prominent causes of trauma fatality [4,5]. Coagulopathy is present in one-quarter of civilian trauma patients and more than quadruples the mortality rate [6,7]. Trauma-induced coagulopathy is a complex disorder including endotheliopathy, platelet dysfunction, decreased clotting factors activity, sympathoadrenal activation, and hyperfibrinolysis [8,9]. Fibrinogen is the final element in the coagulation cascade and a ligand for platelet aggregation. During tissue and vascular injury, it is converted by thrombin to fibrin, an essential part of a blood clot. Of all coagulation factors, fibrinogen is the first to reach significantly low levels in bleeding trauma patients [10]. Hypofibrinogenemia is associated with increased blood loss and a higher number of transfusions and correlates with a worse outcome in injured patients. [11,12,13,14].

Normal fibrinogen blood levels vary and are typically given as 1.8–4.2 g/L. Thus, the absence of normal fibrinogen leads to the disruption of said mechanisms and thus, to bleeding complications. According to the literature data, the severity and pattern of clinical manifestations of bleeding are dependent on the fibrinogen levels [15].

The fifth edition of The European guideline on management of major bleeding and coagulopathy following trauma recommends that monitoring and measures to support coagulation should be initiated immediately upon hospital admission and in a goal-directed manner. In addition, it recognizes early administration of fibrinogen concentrate as an alternative to the high plasma to red blood cells (RBC) ratio transfusion strategy, with a recommended trigger of 1.5 g.L^−1^ [16]. Although this approach is still not commonly accepted, several papers, including the RETIC trial, suggest that it may be superior in reversing coagulopathy or associated with improved patient outcomes [17,18,19,20]. Furthermore, timely identification of hypofibrinogenemia is essential to trigger fibrinogen substitution.

Plasma fibrinogen levels in trauma patients inversely correlate with signs of shock, as shown by the base deficit, low systolic blood pressure, and increased heart rate [21,22,23,24]. Shock index (SI)—a ratio of heart rate to systolic blood pressure, introduced by Allgöwer and Burri in 1967—was shown to foresee mortality, transfusion requirements, and the need for haemostatic resuscitation in trauma as well as after massive obstetric haemorrhage [25,26,27,28,29,30,31,32].

Mutschler et al. detected in the retrospective cohort of 16,305 trauma patients a positive association between SI and the degree of traumatic shock defined by a base deficit value. Further, both thrombocyte count and Quick’s value decreased with the base deficit, while serum lactate and fresh frozen plasma transfusion requirements rose, suggesting that coagulopathy was progressing with shock severity. This was further confirmed by another retrospective database analysis done by the same group, where the increase in SI was associated with increased initial transfusion requirements [28,33].

We decided to perform a prospective observational cohort pilot study evaluating the predictive value of the prehospital and on-admission shock index for low fibrinogen levels in severe trauma patients, with a particular interest in SI ≥ 1, which can be recognized at first sight on the patient’s monitor. We hypothesized that the SI measured in the prehospital setting or on admission to the hospital could predict low initial plasma fibrinogen in trauma patients, thus allowing one to identify trauma patients in need of early targeted fibrinogen substitution.

## 2. Materials and Methods

We performed a prospective observational cohort pilot study in two level 1 trauma centres in Ústí nad Labem and Plzeň, Czech Republic, covering a population of 1.4 million people. All patients aged > 18 years meeting the national triage criteria for severe trauma (Table 1) and managed on-scene by the helicopter emergency medical service (HEMS) admitted between January 2016 and February 2017 were eligible for the study. Patients known to have factors affecting fibrinogen level (pregnancy, malignant or inflammatory disease), those who had received fresh frozen plasma or fibrinogen concentrate before blood sampling, or whose printed prehospital vital signs measurement was not available, were retrospectively excluded from the study. Consistently with the most recent guidelines, hypofibrinogenemia was defined as plasma fibrinogen level of 1.5 g.L^−1^ or less [12]. This value was used for further analyses.

Prehospital management of the patients followed the contemporary guidelines for treatment of major bleeding in trauma patients [34]. Noninvasive blood pressure using the oscillometric method and heart rate from electrocardiography were measured by the Emergency Medical System (EMS) crew using Lifepak 15 (Physio-Control, Inc., Redmond, WA, USA) or Corpuls 3 (GS Elektromedizinische Geräte G. Stemple GmbH, Kaufering, Germany) monitors as soon as possible after the arrival at the scene and repeated as needed. No blood products were administered to patients in the field. After the patient’s handover in the emergency room, blood for coagulation testing was sampled into plastic tubes containing 0.129 M trisodium citrate (BD Vacutainer, Plymouth, UK) and transported to a local laboratory. Both centres determined fibrinogen plasma concentration using the Clauss method (Sysmex CS 5100, Siemens Healthcare GmbH, Erlagen, Germany).

Demographic data, time course from the injury to blood sampling, laboratory values (fibrinogen concentration), clinical findings (all EMS and first hospital blood pressure and heart rate measurements), Injury Severity Score (ISS), and circumstances (type of trauma and transport) were prospectively collected in all cases and archived for further evaluation. SI was calculated as the ratio between systolic blood pressure and heart rate from all available pairs of these values using MS Excel (Microsoft Corp., Redmond, WA, USA). The highest value during EMS care defined the worst prehospital SI. Both the worst-prehospital and the first on-admission SI values were used for further analysis. As it is clinically simple to identify patients with SI ≥ 1 (heart rate is equal to or higher than the systolic blood pressure), we used it as the cut-off value for calculations. Further, SI with the best predictive value was estimated using Youden’s J.

Data are presented as mean +/− standard deviation or median and interquartile range (IQR) for continuous variables. Percentages are used for categorical variables. Continuous variables were tested for normal distribution using the Kolmogorov–Smirnov test. Student’s *t*-test, or the Mann–Whitney U test, was used to detect differences between groups as appropriate. For categorical variables, the Chi-squared test was used to compare groups. To test the diagnostic performance, receiver operating characteristics curves (ROC) were constructed to cut off fibrinogen concentration 1.5 g.L^−1^. The area under the curve (AUROC), sensitivity, specificity, positive predictive value (PPV), and negative predictive value (NPV) were calculated for SI ≥ 1. STATISTICA 13.2 (TIBCO Software Inc., Palo Alto, CA, USA) was used for statistical analysis and testing. Origin Pro 9.1 (OriginLab, Northampton, MA, USA) was used for ROC analysis. The sample size was calculated using the online calculator (www.openepi.com, accessed on 15 September 2015). Based on prior clinical experience, we assumed that 10% of patients have hypofibrinogenemia on admission to hospital. A two-sided confidence level of 95% and a power of 80% were used for calculation, giving a sample size of 300 patients.

The study was registered: Clinicaltrials.gov ID: NCT0269539 (Registered 27 November 2015).

## 3. Results

Three hundred and twenty-two patients were assessed for eligibility, and 318 met the inclusion criteria. Of these, 264 (83%) were included for further analysis, as detailed in Figure 1. The reason for exclusion was the absence of printed prehospital vital signs records (22, 6.8%), missing fibrinogen on-admission level (11, 3.4%), or known malignancy or inflammatory disease (21, 6.5%). The general characteristics of enrolled patients are reviewed in Table 2. Several differences in injury severity and SI values between the two centres were identified.

The shock index ≥ 1 was found in 36 (13.7%) patients during the prehospital course and in 17 (6.5%) patients on admission to hospital. Eleven (4.2%) patients had plasma fibrinogen concentration lower than 1.5 g.L^−1^. The area under the receiver operating characteristics curve (AUROC) for plasma fibrinogen < 1.5 g.L^−1^ was 0.79 (95% CI 0.64–0.91) for the worst prehospital SI and 0.77 (95% CI 0.64–0.91) for the on-admission values (Figure 2). The best Youden’s index was 0.54 for the worst prehospital SI value of 0.92 and 0.63 for SI value of 0.8 on-admission.

The value of SI ≥ 1 indicated severe hypofibrinogenemia with the negative predictive value of 0.98 (95% CI 0.96–0.99) for worst prehospital and 0.97 (95% CI 0.95–0.98) for on-admission data. Positive predictive value was 0.15 (95% CI 0.08–0.27) for prehospital SI and 0.19 (95% CI 0.07–0.41) for admission data. Detailed validity scores calculated for SI ≥ 1 and SI with best Youden’s index are presented in Table 3.

Shock index values < 1 excluded critical fibrinogen levels below 1.5 g.L^−1^ with 97% prediction. The accuracy of SI in detecting plasma fibrinogen levels recommended to trigger substitution was 87% and 92%, respectively. The positive predictive value of SI was weak and not exceeding 19% for <1.5 g.L^−1^.

## 4. Discussion

The main finding of the presented study is that both worst prehospital and on-admission SI showed an excellent negative predictive value of 97% for critically low plasma fibrinogen in the first hospital blood sample. The positive predictive value was low and did not exceed 19%. We hypothesise that factors such as anxiety or pain suffered by the patient might have raised the heart rate and moved SI towards higher values without necessarily being related to the presence of traumatic shock. Predictive values of SI ≥ 1 and SI ≥ value with the best Youden’s index were comparable, indicating that simply comparing a patient’s heart rate with systolic blood pressure has a similar performance as the exact computation of the shock index.

Early and targeted management of trauma-induced coagulopathy with coagulation factor concentrates appears to reduce blood product requirements and the incidence of multi-organ failure and overcomes logistic problems related to fresh frozen plasma or cryoprecipitate use. Fibrinogen concentrate also has a favourable safety profile and standardised fibrinogen content [18,35]. This strategy is increasingly adopted during targeted hospital management of major trauma bleeding patients and received interest in the pre-emptive or even prehospital settings as well [19]. The FIinTIC pilot study has shown that prehospital administration of fibrinogen concentrate in patients with major trauma bleeding is associated with increased clot firmness on admission as assessed by a viscoelastic test (FIBTEM) [36].

The optimal threshold for fibrinogen substitution and target plasma concentration remains poorly defined. Laboratory model suggests that clot formation is reduced below 2.0 g.L^−1^. Further, Hagemo found that levels below 2.29 g.L^−1^ (although still in the normal range) correlate with worse outcomes [13]. Levels below 1.5 g.L^−1^ are recommended to trigger fibrinogen substitution by the current guidelines [16].

Plasma fibrinogen level is usually determined using the Clauss assay via a central laboratory which is time-consuming and may take 45 min or more [22]. Viscoelastic testing has been introduced as a point-of-care method and overcomes the long laboratory turnaround times. Viscoelastic haemostatic assays (VHAs) are whole blood point-of-care tests that have become an essential method for assaying haemostatic competence in liver transplantation, cardiac surgery, and most recently, trauma surgery involving haemorrhagic shock [37]. However, it typically requires a dedicated operator or may not be available in a resource-limited environment. These factors may limit the widespread use of the method, delay recognition of hypofibrinogenemia, and make it unavailable in the prehospital setting. Thus, a quick, reliable and easy-to-perform detection tool is clinically relevant. 

Regarding the potential of prehospital fibrinogen concentrate use, a clear identification of patients at risk of hypofibrinogenemia remains challenging [38]. From the technical point of view, the ideal test should be easy to perform and interpret and should not prolong the time elapsed between injury and bleeding control. It should rely only on technology in the ambulance or aircraft and be robust enough to sustain the environment (i.e., wide ambient temperature and humidity operating range). No such point-of-care technology is currently available. Shock index can be assessed without needing equipment other than a tonometer and a clock and may overcome technical issues of the more sophisticated methods.

To our best knowledge, this study is the first one directly evaluating the association between SI and plasma fibrinogen level on admission. Detection of hypofibrinogenemia is crucial, as such information should trigger an immediate clinical and targeted response [16]. Shock index is an easy-to-use parameter as it requires only information that is ubiquitous in the emergency medicine environment—a patient’s heart rate and systolic blood pressure. Cut-off level of 1 is easily assessed by simple comparison of the values displayed on each patient monitor. Thus, SI might be a tool allowing for early identification of patients at risk of hypofibrinogenemia, even in the prehospital setting, where no other tool to assess the risk of hypofibrinogenemia has been introduced to date.

Several other studies evaluated the use of SI as the prediction tool in the early trauma course. Paladino et al. found an isolated on-admission SI value to have poor discrimination capability between a minor and major trauma. The calculated area under the ROC curve of 0.63 resulted in a major injury miss in 44% of cases [39]. In a recent paper, Costa et al. found that prehospital SI 0.9 predicted the need for a massive transfusion with 90.5% specificity and 0.71 (CI 95% 0.64–0.73) area under the ROC curve. This study was based on a large cohort of 13,222 major trauma patients from the Swiss Trauma Registry. These findings are in line with the results of our study, where we found 88% specificity and AUROC 0.76 (CI 95% 0.64–0.86) for the prehospital SI ≥ 1 and hypofibrinogenemia < 1.5 g.L^−1^ [40].

Other studies focused on the early identification of hypofibrinogenemia. Schlimp reported that fibrinogen level upon admission correlated with haemoglobin, base excess, and ISS [22]. Tonglet introduced a Trauma-Induced Coagulopathy Clinical Score (TICSS) comprising general trauma severity, the extent of injuries, and systolic blood pressure to predict the need for damage-control resuscitation (DCR) in blunt trauma. A cut-off of 10 points provided a PPV of 72.7% for the need for DCR, including the low fibrinogen on admission [41]. David found in a retrospective trial that vasopressor administration, ISS, shock index, and fluid volume were independent predictors of trauma-induced coagulopathy (defined as fibrinogen < 1.5 g.L^−1^ or prothrombin time ratio > 1.5 or platelet count < 100 × 10^−9^.L^−1^) [42]. Finally, Gauss derived a Fibrinogen in Trauma (FibAT) score of 13 different clinical and laboratory findings, with a maximum accuracy of 86% [24]. While the performance of these tests was like the one found for SI in our study, they are typically based on multiple parameters and require laboratory tests or a CT scan. Compared to this, SI is much faster and easier to handle in the hasty environment of trauma care.

This study has several limitations. First, only trauma patients admitted from the helicopter emergency medical service were enrolled. Convenience sampling was chosen, as we preferred the prehospital vital signs printout from the patient monitor over the hand-written values in the patient chart. Accepting only monitor printouts allowed us to exclude bias from incorrect or missing vital signs values. Compared to the ground ambulance service, HEMS staff is limited in its number, allowing for better compliance with printout handover. As AUROC for hypofibrinogenemia detection by the SI in our study was similar to the one for massive transfusion prediction in the study of Costa et al., we hypothesise that the bias introduced by the convenience sampling method is not significant for the interpretation of the results [40].

Second, prehospital shock index values were significantly higher in one of the centres (Plzeň), as was the proportion of patients with SI ≥ 1 and higher mean ISS. This may reflect the difference in trauma triage thresholds between the two regions and is also mirrored in the trend towards lower fibrinogen in this group, suggesting a dose–response relationship between the degree of shock during the prehospital phase and the level of hypofibrinogenemia. Interestingly, the difference in shock index disappeared in on-admission values, indicating that hemodynamic status alone is not the only contributor to hypofibrinogenemia that persists despite volume resuscitation. This is in line with the concept of trauma-induced coagulopathy [8,9].

Third, only a tiny proportion of patients (4.2%) had hypofibrinogenemia < 1.5 g.L^−1^. We consider this to be a combined effect of a low mean ISS (13.6) and a short mean time from injury to admission (68 min). The small number of patients with hypofibrinogenemia likely affected the positive predictive value of SI ≥ 1, which proved to be very low (16% for prehospital and 19% for on-admission SI). We suggest that the results of this pilot study should be validated on a more extensive database.

Fourth, we did not consider any interventions possibly undertaken during the prehospital treatment. For example, haemodilution by the volume replacement therapy might have affected fibrinogenaemia, and vasopressor infusion could have influenced the systolic blood pressure or heart rate used for calculating the SI. However, the prehospital treatment of trauma patients followed the valid guidelines prioritizing swift transport and implementing the permissive hypotension strategy in the absence of brain injury [34]. Therefore, with the short time from injury to admission, we do not consider this an essential source of bias. Notably, this study aimed to evaluate the SI as a simple and real-life accessible tool. From this point of view, passing over the potential effect of prehospital treatment appears to be a strength rather than a limitation, as it mirrors the real potential use of SI in the emergency setting.

Fifth, 98% of patients in this cohort suffered blunt trauma, and the study was performed in a civilian setting. Therefore, results may not reflect the context of penetrating injuries or combat casualties.

## 5. Conclusions

Shock index is an easy-to-perform clinical tool which may help identify adult trauma patients at risk of hypofibrinogenemia from very early contact. Shock index values < 1 excluded severe hypofibrinogenemia with 97% predictive value. Taken to the patient’s bedside, this means that the risk of severe hypofibrinogenemia was low whenever the systolic blood pressure remained higher than the heart rate. These findings should raise interest in the use of this simple clinical tool. Validation of the shock index in detecting hypofibrinogenemia on a larger trauma patient database is warranted.

## Figures and Tables

**Figure 1 jcm-12-01707-f001:**
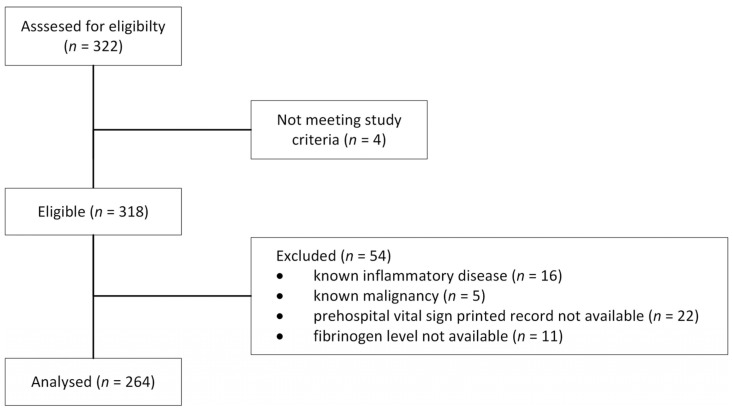
Enrolment flow diagram.

**Figure 2 jcm-12-01707-f002:**
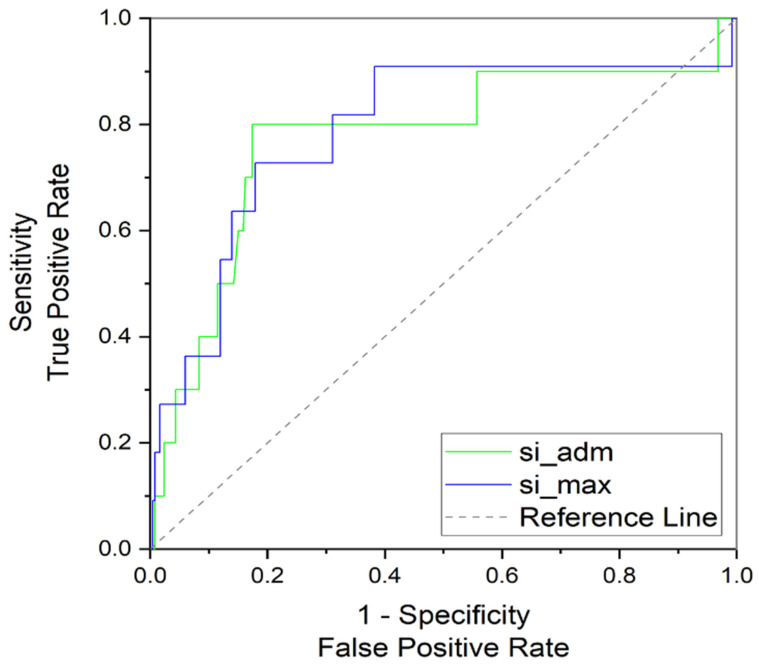
Receiver operating characteristics curve for severe hypofibrinogenemia and worst prehospital (si_max) and admission (si_adm) shock index.

**Table 1 jcm-12-01707-t001:** The national criteria for on-scene trauma triage.

Physiological criteria	Respiratory rate of <10 or >29 breaths per minute Systolic blood pressure < 90 mmHg Glasgow Coma Scale < 13 points Persisting focal neurological deficit
Anatomical criteria	Penetrating head trauma Chest wall instability or deformity Penetrating chest trauma Penetrating abdominal trauma Pelvic instability 2 or more long-bone fractures (femur, humerus, tibia)
Mechanism of injury	Falls from a height of >6 m Vehicle run over Pedestrian struck > 35 km.h^−1^ Ejection from vehicle Intrusion in vehicle Death in the same passenger compartment Burial Another mechanism with the equivalent energy
Auxiliary criteria	Age ≤ 6 or ≥ 60 years Major cardiopulmonary comorbidity Intoxication

**Table 2 jcm-12-01707-t002:** Study population demographic and clinical characteristics. Data are presented as number (%), mean (SD) or median (IQR) values.

	All	Centre 1 (Ustí nad Labem)	Centre 2 (Plzen)	*p* Value ^1^
n (%)	264 (100)	155 (59)	109 (41)	N/A
Age (SD)	42.1 (15.4)	41.2 (15.2)	43.4 (15.5)	0.23
Male (%)	196 (74)	112 (72)	84 (77)	0.37
Injury Severity Score (SD)	13.6 (11.8)	11.2 (10.6)	16.9 (12.7)	<0.001
Injury Severity Score ≥ 16 (%)	86 (32.6)	40 (25.8)	46 (42.2)	0.005
Blunt trauma (%)	260 (98)	153 (99)	107 (98)	0.72
Prehospital time [min] (SD)	68 (20.7)	63 (16.7)	76.3 (23.9)	<0.001
Time to sample [min] (SD)	76 (21.7)	70.5 (18.8)	85.1 (23.2)	0.02
Prehospital SI (IQR)	0.70 [0.59–0.86]	0.69 [0.59–0.83]	0.78 [0.61–0.93]	0.01
Admission SI (IQR)	0.64 [0.54–0.76]	0.65 [0.54–0.78]	0.63 [0.54–0.76]	0.48
Fibrinogen, g.L^−1^ (IQR)	2.69 [2.29–3.12]	2.70 [2.31–3.25]	2.61 [2.27–2.98]	0.13
Fibrinogen <2.3 g.L^−1^ (%)	69 (26.1)	38 (24.5)	31 (28.4)	0.47
Fibrinogen <2.0 g.L^−1^ (%)	32 (12.1)	16 (10.3)	16 (10.3)	0.29
Fibrinogen <1.5 g.L^−1^ (%)	11 (4.2)	7 (4.5)	4 (3.7)	0.73
Prehospital SI ≥ 1 (%) *	36 (13.7)	12 (7.8)	24 (22)	<0.001
Admission SI ≥ 1 (%) *	17 (6.5)	8 (5.2)	9 (8.3)	0.30

* N/A, not applicable; SI, Shock Index; ^1^ difference between centres.

**Table 3 jcm-12-01707-t003:** Results—Performance of prehospital and on-admission shock index (SI ≥ 1 and SI ≥ value with the best Youden’s index) in the detection of hypofibrinogenemia <1.5 g.L^−1^.

Shock Index	Sensitivity (95% CI)	Specificity (95% CI)	PPV (95% CI)	NPV (95% CI)	Accuracy (95% CI)
Prehospital SI ≥ 1	0.50 (0.19–0.81)	0.88 (0.83–0.92)	0.15 (0.08–0.27)	0.98 (0.96–0.99)	0.87 (0.82–0.90)
Prehospital SI ≥ 0.92	0.73 (0.39–0.94)	0.82 (0.76–0.86)	0.15 (0.11–0.26)	0.99 (0.99–0.99)	0.81 (0.76–0.86)
Admission SI ≥ 1	0.30 (0.07–0.65)	0.94 (0.91–0.97)	0.19 (0.07–0.41)	0.97 (0.95–0.98)	0.92 (0.88–0.95)
Admission SI ≥ 0.8	0.80 (0.44–0.97)	0.80 (0.74–0.85)	0.15 (0.11–0.25)	0.99 (0.99–1.0)	0.80 (0.75–0.85)

## Data Availability

The data presented in this study are available on request from the corresponding author. The data are not publicly available due to privacy policy.

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
