# Peer review of "Shock Index for Early Detection of Low Plasma Fibrinogen in Trauma: A Prospective Observational Cohort Pilot Study"

_jcm, 2023, doi:10.3390/jcm12041707_

Round 1
Reviewer 1 Report
The authors describe a very interesting original data, which is focused Shock index for early detection hypofibrinogenemia in trauma. The Shock Index is a hemorrhage indicator with a cut-off point for the risk of bleeding
The scope of the manuscript is reasonable, the authors do well in staying focused and on key point. The authors do a solid job. Text is supported by several charts/tables and figures, which substantively add to the manuscript.
Major comments:
- Page 2 lines 47-48: The authors forgot to mention the physiological concentration of fibrinogen in the blood. ,,Normal fibrinogen blood levels vary and are typically given as 1.8–4.2 g/L.“ At the same time, the following should be added to the manuscript: ,, Thus, the absence of normal fibrinogen leads to the disruption of said mechanisms and thus to bleeding complications. According to the literature data, the severity and pattern of clinical manifestations of bleeding are dependent on the fibrinogen levels“ It is appropriate to mention the manuscript in which those facts were written: J. Clin. Med. 2022, 11(4), 1083; https://doi.org/10.3390/jcm11041083
- Page 7 lines 197-204 Authors should include important facts about VHA in the manuscript: ,,Viscoelastic hemostatic assay (VHAs) are whole blood point-of-care tests that have become an essential method for assaying hemostatic competence in liver transplantation, cardiac surgery, and most recently, trauma surgery involving hemorrhagic shock. At the same time, they should cite the manuscript in which this statement was published:“ J. Clin. Med. 2022, 11(3), 860; https://doi.org/10.3390/jcm11030860
I have to say that with these 34 references there very few references from the last 5 years. This is insufficient, authors should use newer references.
Author Response
Dear Sir/Madame,
We sincerely appreciate the insights and comments you have made to the article we submitted to this journal.
We agree with the amendments you suggested, and both were added to the appropriate sections of the manuscript (lines 49-52 and 205-207). Also, relevant references were added to the list.
Further, we have updated the reference list by the following papers:
Eastridge BJ, Holcomb JB, Shackelford S. Outcomes of traumatic hemorrhagic shock and the epidemiology of preventable death from injury. Transfusion (Paris) 2019; 59: 1423–1428.
Candefjord S, Asker L, Caragounis EC. Mortality of trauma patients treated at trauma centers compared to non-trauma centers in Sweden: a retrospective study. European Journal of Trauma and Emergency Surgery 2022; 48: 525–536.
Grottke O, Mallaiah S, Karkouti K, et al. Fibrinogen Supplementation and Its Indications. Semin Thromb Hemost 2020; 46: 38–49.
James A, Abback PS, Pasquier P, et al. The conundrum of the definition of haemorrhagic shock: a pragmatic exploration based on a scoping review, experts’ survey and a cohort analysis. European Journal of Trauma and Emergency Surgery 2022; 48: 4639–4649.
Ziegler B, Bachler M, Haberfellner H, et al. Efficacy of prehospital administration of fibrinogen concentrate in trauma patients bleeding or presumed to bleed (FIinTIC): A multicentre, double-blind, placebo-controlled, randomised pilot study. European Journal of Anaesthesiology 2021; 38: 348–357.
Volod O, Bunch CM, Zackariya N, et al. Viscoelastic Hemostatic Assays: A Primer on Legacy and New Generation Devices. Journal of Clinical Medicine; 11. Epub ahead of print 1 February 2022. DOI: 10.3390/jcm11030860.
Costa A, Carron P-N, Zingg T, et al. Early identification of bleeding in trauma patients: external validation of traumatic bleeding scores in the Swiss Trauma Registry. Crit Care 2022; 26: 296.
Brunclikova M, Simurda T, Zolkova J, et al. Heterogeneity of Genotype–Phenotype in Congenital Hypofibrinogenemia—A Review of Case Reports Associated with Bleeding and Thrombosis. Journal of Clinical Medicine; 11. Epub ahead of print 1 February 2022. DOI: 10.3390/jcm11041083.
Kind regards,
Josef Skola
on behalf of the author group

Reviewer 2 Report
It is an honor to review your manuscript.
The design of the manuscript is very good. However, since the shock index is an indicator using only vital signs, the usefulness of predicting fibrinogen is questionable.
A high SI means that the possibility of mortality or massive blood transfusion is high, so fibrinogen is naturally low.
In order for a paper to be published, the introduction and discussion should provide a more attractive description of the usefulness of SI to predict low fibrinogen.
Or, it seems that the authors should use a method to show the usefulness and novelty of the paper by adding an outcome variable other than fibrinogen.
Author Response
Dear Sir/Madame,
Thank you for reading our work and providing valuable insight into it. It was much appreciated. Following your recommendation, we have updated the text, better explaining the potential beneficial use of the shock index in the detection of hypofibrinogenemia in trauma patients (lines 81-82 and 226-231).
Kind regards,
Josef Skola
on behalf of the author group.
Round 2
Reviewer 1 Report
The presented manuscript has been corrected in response to the suggestions. The authors have followed the recommendations of the reviewer. After the revision, the provided data and addition of the results became more clear. I would like to thank the authors for resubmitting the manuscript and explaining the obscure points from the previous version.
Reviewer 2 Report
Thank you for your hard work on editing.